# Upcycling rice yield trial data using a weather-driven crop growth model

Hiroyuki Shimono [1,2✉], Akira Abe [3], Chyon Hae Kim[4], Chikashi Sato[5] & Hiroyoshi Iwata[6]

Efficient plant breeding plays a significant role in increasing crop yields and attaining food security under climate change. Screening new cultivars through yield trials in multi-environments has improved crop yields, but the accumulated data from these trials has not been effectively upcycled. We propose a simple method that quantifies cultivar-specific productivity characteristics using two regression coefficients: yield-ability ($\beta$) and yield-plasticity ($\alpha$). The recorded yields of each cultivar are expressed as a unique linear regression in response to the theoretical potential yield ($Y_p$) calculated by a weather-driven crop growth model, called as the "YpCGM method". We apply this to 72510 independent datasets from yield trials of rice that used 237 cultivars measured at 110 locations in Japan over 38 years. The YpCGM method can upcycle accumulated yield data for use in genetic-gain analysis and genome-wide-association studies to guide future breeding programs for developing new cultivars suitable for the world's changing climate.

[1] Faculty of Agriculture, Iwate University, Morioka, Iwate 020-8550, Japan. [2] Agri-Innovation Center, Iwate University, Morioka, Iwate 020-8550, Japan. [3] Iwate Biotechnology Research Center, Kitakami, Iwate 024-0003, Japan. [4] Faculty of Science and Engineering, Iwate University, Morioka, Iwate 020-8550, Japan. [5] Ifuu Rinrin, 77-9, Rikuzentakata, Iwate 029-2205, Japan. [6] Laboratory of Biometry and Bioinformatics, University of Tokyo, Bunkyo-ku, Tokyo 113-8657, Japan. ✉email: shimn@iwate-u.ac.jp

Crop yield must improve to ensure food security by meeting the growing demand for food, which is predicted to increase by 70% by 2050[1]; however, this challenge must be accomplished under the constraints imposed by future climate change. Genetic improvement of crop performance will be one important solution. Recent advances in genotyping technologies have accelerated our ability to screen candidate high-yielding cultivars efficiently using methods such as genome-wide association studies (GWAS) and genomic prediction (GP)[2–10]. Although genotyping a given cultivar is now relatively easy and affordable using genome-wide DNA sequencing, phenotypic analyses based on crop yield must be repeatedly evaluated in season-long field trials across a range of environments. The yield of a given cultivar is not stable across environments since yield results from the interactions of many physiological processes that respond to environmental fluctuations throughout the growing season. Thus, strong genotype by environment (G × E) interactions make it difficult to predict crop yields across a range of environments.

The pioneering study of Finlay and Wilkinson[11] proposed a method for quantifying the G × E interaction by linear regression (the FW method). The FW method standardized the observed yield of each cultivar in a given environment against a mean yield of all cultivars in a comparable environment (Supplementary Fig. S1a). The slope of the regression of the observed yield as a function of the mean yield represents the plasticity of the yield response, with the average plasticity equal to 1.0; cultivars with above-average plasticity have slopes >1.0, and cultivars with below-average plasticity have slopes <1.0. This method represents a foundational achievement for evaluating G × E interactions and has been widely used in plant science, including for genomic analyses[12–18]. However, the FW method can only be used to compare cultivars with yield data measured at the same site and season in a side-by-side yield comparison, in which different cultivars grow together. The plasticities of a cultivar determined under independent experiments, not side-by-side yield comparisons, are consequently not comparable.

To solve this problem, we designed this study to develop a new method for characterizing the cultivar-specific yield response standardized using the theoretical potential yield ($Y_p$) by using a weather-driven crop growth model (CGM). We call this the 'YpCGM method' (Supplementary Fig. S1b). The novelty of this method is that it lets researchers combine yield trial data from different studies to estimate $Y_p$ from weather data as inputs for the CGM. Only two parameters are required: the yield-plasticity (α; dimensionless) and the yield-ability (β; t/ha). These values represent the slope and estimated yield, respectively, at the standardized potential yield (SPY; t/ha) in a cultivar-specific regression (described in the Methods section). The regression provides a simple expression of a cultivar's yield characteristics (α and β). This analysis differs from previous modeling studies that required the measurement and parameterization of many physiological processes for each cultivar, including leaf expansion, photosynthesis, biomass production, and carbon allocation to harvestable organs[19,20]. Our method can be applied in a 'big data' context using the accumulated yield data from many previous studies, including studies that recorded only yield without measuring the wide range of physiological processes required to parameterize a CGM. Reuse (i.e., upcycling) of these data for yield phenotyping has great potential to help breeders identify promising cultivars that will permit them to boost crop yields.

We applied this method to datasets from 72,510 yield trials under similar management practices for disease and pest control with 237 core cultivars of rice (*Oryza sativa* L.) evaluated from 1980 to 2017 at 110 public agricultural experimental stations in Japan along a climatic transect from south to north and from west (facing the Sea of Japan) to east (facing the Pacific Ocean), covering the latitudes from 30°N to 43°N (Supplementary Fig. S2). We validated the reliability of our method of calculating $Y_p$ using 10 replications of 10-fold cross-validation for the cultivars based on their pedigree and genome. Cultivar-specific parameters were used to analyze the genetic gain, and GWAS can be used to evaluate future cultivars suitable for the world's changing climate.

## Results

**$Y_p$ captures cultivar-specific yield characteristics.** The phenotype data for the 237 cultivars were recorded from 20 to 6342 yield trials under a range of environmental conditions. For example, the popular cultivar 'Koshihikari', which was a parent of 853 progeny, provided phenotypic data from 6342 trials, with yields that ranged from 0.9 to 8.7 t/ha (Supplementary Fig. S6vii). Days to heading ranged from 39 to 117 (Supplementary Fig. S6xvii), panicles per m² ranged from 178 to 684 (Supplementary Fig. S6xxvii), and panicle length ranged from 12.6 to 23.8 cm (Supplementary Fig. S6xxxvii). The other nine cultivars in Supplementary Fig. S6i–xl, which currently account for more than 70% of the rice production in Japan, showed a similar range of variation in their phenotypes. Data for all 237 cultivars are listed in Supplementary Data 7 and 8.

To quantify the genotypic coefficients of yield-ability (β in equation 1, defined as the expected yield at $Y_p = 8$ t/ha SPY) of a given cultivar and its plasticity (α) using the YpCGM method, we plotted the observed yields per cultivar against $Y_p$ estimated from weather records by the weather-driven CGM. Prediction accuracy was evaluated by comparison with the cumulative air temperature and solar radiation and with the observed panicle number and the observed panicle length, two characteristics that strongly determine yield variation in rice[21]. The *RMSE* of the difference between the observed yield and the yield predicted by $Y_p$ ranged from 0.36 to 1.45 t/ha and averaged 0.84 t/ha for the 237 cultivars (Supplementary Data 8, Fig. S7i–x). To evaluate the accuracy, the *RMSE* of the yield predicted by $Y_p$ was plotted against the *RMSE* predicted by the four variables (Supplementary Fig. S8). *RMSE* predicted by $Y_p$ was 3.8% to 4.2% smaller than that predicted by the cumulative air temperature (Supplementary Fig. S8a), the cumulative solar radiation (Supplementary Fig. S8b), and the observed panicle length (Supplementary Fig. S8c and Fig. S7xxi–xxx). The *RMSE* predicted by $Y_p$ was lower than that of all four variables, and it was notable 1.2% lower than that predicted by the observed panicle number, which is key driver of variation of rice yield to environments[21] (Supplementary Fig. S8d and Fig. S7xi–xx). On this basis, $Y_p$ appears to be an appropriate index that accounts for the variation of the observed yield of each cultivar in response to environmental changes.

The genotypic β coefficient ranged from 2.5 to 7.3 t/ha among the 237 cultivars (Fig. 1a and Supplementary Data 1), and the α coefficient ranged from −0.23 to +0.95 (Fig. 1b, Supplementary Data 1). Figure S7i–x is an example of how the coefficients were quantified for the 10 major cultivars by YpCGM. Supplementary Data 7 lists the coefficients of all 237 cultivars. The robustness of the regression models estimated for each cultivar was evaluated by leave-one-out cross-validation. As a result, the difference between the predicted and the fitted coefficients of determination was small for all cultivars (Supplementary Fig. S9), suggesting that the regression model does not tend to overfit and is robust in prediction. Note that there were cultivars in which the coefficient of determination for prediction was lower than the coefficient of determination for fitting, but these were those for which the number of environments (number of data used in the regression) was less than 50. In other words, as the number of data increases, the robustness of the regression model is expected to improve further.

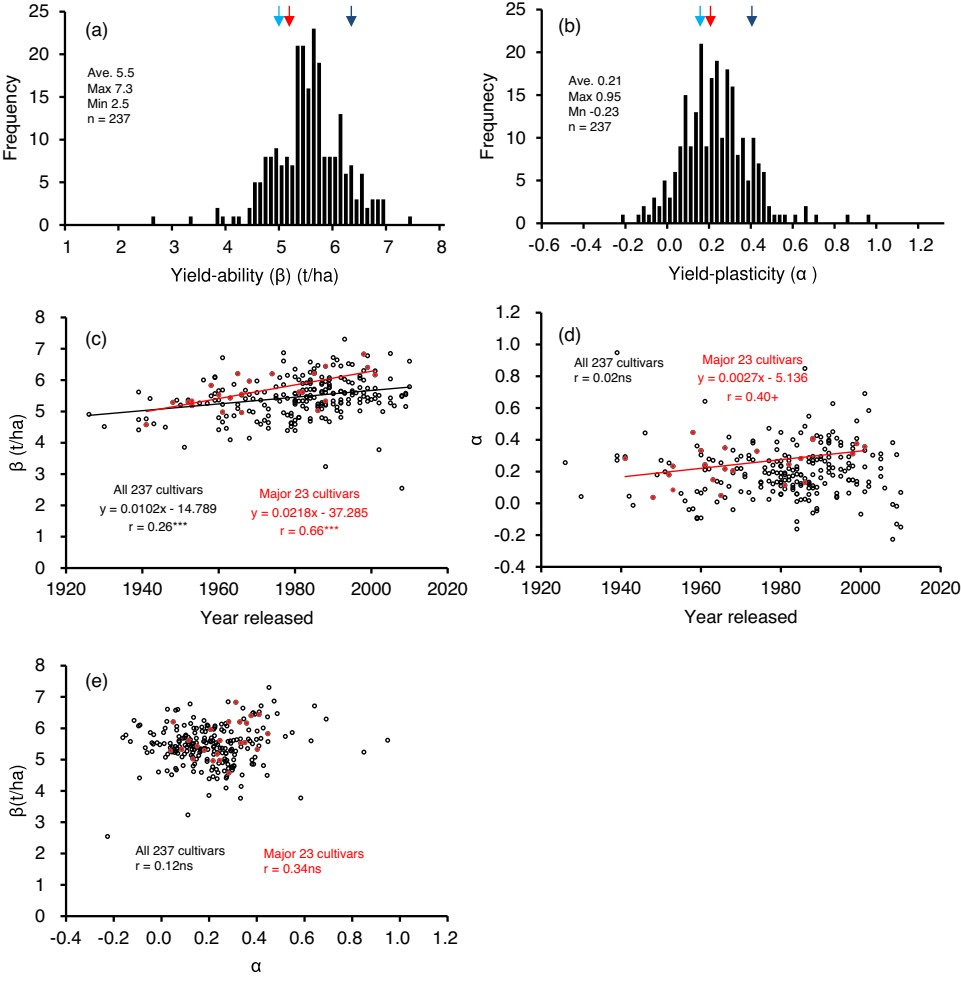

**Fig. 1 Genotypic coefficients for yield-ability (β) and yield-plasticity (α) of the 237 core cultivars.** Frequency distributions for (**a**) yield-ability and (**b**) yield-plasticity. Relationships between the year when a cultivar was released and the (**c**) yield-ability and (**d**) yield-plasticity. **e** Relationship between yield-ability and yield-plasticity. Red, light blue, and dark blue arrows in (**a**) and (**b**) represent 'Koshihikari', a long-time popular cultivar; 'Nipponbare', an older popular cultivar that was used for the rice genome project; and 'Hokuriku 193', a high-yielding cultivar, respectively. Black and red data points in (**c**), (**d**), and (**e**) indicate data for all 237 cultivars and for the 23 major cultivars that represent the most widely grown cultivars from 1920 to 2020 (i.e., farmer favorite historical cultivars), respectively. ***$P < 0.001$, +$P < 0.1$, ns not significant. The 23 major cultivars were 'Norin 22' (1941), 'Kimmaze' (1948), 'Hounenwase' (1952), 'Koshihikari' (1953), 'Etsujiwase' (1953), 'Fujiminori' (1958), 'Sasanishiki' (1960), 'Nipponbare' (1961), 'Reimei' (1963), 'Todorokiwase' (1965), 'Toyonishiki' (1966), 'Reiho' (1966), 'Ishihikari' (1968), 'Akihikari' (1974), 'Yukihikari' (1981), 'Akitakomachi' (1982), 'Kirara 397' (1985), 'Hinohikari' (1986), 'Hitomebore' (1988), 'Fukuhibiki' (1988), 'Nanatsuboshi' (1998), 'Masshigura' (1999), 'Hokuriku 193' (2001). The numbers in parentheses are the release years.

The relationships of α and β to the observed values of mean yield, panicle number, and panicle length for each cultivar are examined in Supplementary Fig. S10. Both α and β were significantly positively correlated with mean observed yield (Supplementary Fig. S10a, d) although the correlation coefficient of α was lower. Interestingly, both coefficients were correlated significantly negatively with the mean observed panicle length (Supplementary Fig. S10b, e) and significantly positively with the mean observed panicle number (Supplementary Fig. S10c, f).

**GP for explaining variations in the genotypic coefficients.** Using 10-fold cross-validation with 10 replicates, genomic prediction explained the variation of β based on genomic information from 91,800 SNPs, with a significant positive correlation ($r = 0.562 \pm 0.012$) in the gBLUP, which was similar to the result for the pBLUP ($r = 0.481 \pm 0.009$) and the gpBLUP ($r = 0.571 \pm 0.008$) and the g×pBLUP ($r = 0.576 \pm 0.008$) for β (Fig. 2a and Supplementary Data 2). The α was predicted with similar accuracy by the four types of BLUPs, at $r = 0.178$ to 0.211,

and the values were lower than the correlations for β. Interestingly, the *b* parameter in the basic form of equation 1 that does not include *SPY* was not explained by the genome and pedigree information ($r = -0.133$ to $-0.054$) (Fig. 2b and Supplementary Data 3). This result demonstrates the effectiveness of using β rather than *b* for characterizing genotypic characteristics.

The heritability of β was significant and high; the heritability was 0.740 for genomic information and 0.745 for pedigree information (Fig. 2c). These values were much higher than the heritability values for α, at 0.326 and 0.408, respectively. Heritability for *b* was also poor, at 0.181 and 0.278, respectively (Fig. 2d).

**Genetic improvements in Japanese rice breeding since 1920.** Genetic gain is defined as the yield increase over time and can be measured by evaluating the yield performance of rice cultivars released in different years but grown under the same experimental conditions[22,23]. Our novel method let us compare the genetic gain in studies conducted in different years and at

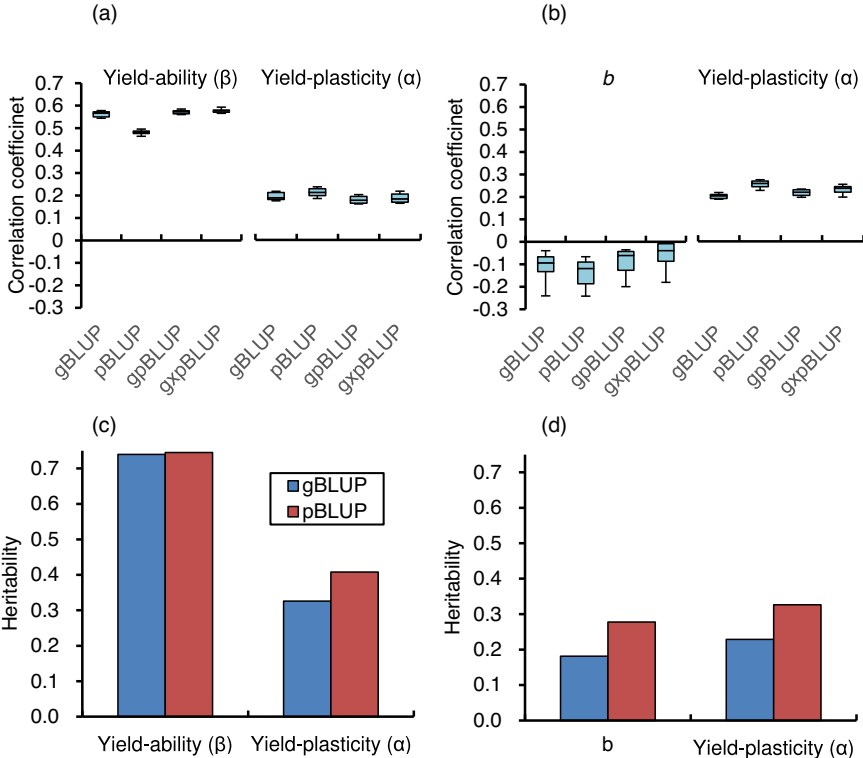

**Fig. 2 Genomic prediction and heritability of yield-ability (β) and yield-plasticity (α) of the 237 core cultivars.** Comparison of the strength to predict (**a**) yield-ability (β) and yield-plasticity (α), and (**b**) b and α by pedigree-based best linear unbiased prediction (pBLUP), genome-based best linear unbiased prediction (gBLUP), their combination (gpBLUP), and genome × pedigree information (g×pBLUP). The correlations between the observed and predicted values were based on a 10-fold cross-validation with 10 replicates. A comparison of the yield heritability for using (**c**) β and α, (**d**) b and α of the 237 core rice cultivars.

different locations with Japan's 237 core cultivars by calculating just two genetic coefficients (α and β). We also analysed the 23 most widely grown cultivars from 1920 to the present (i.e., farmer favorites; Supplementary Fig. S11). The cultivars that we included in our analysis represent a diverse range of maturity groups whose productivity could not be easily compared directly side by side, particularly owing to their differences in photoperiod sensitivity[24].

Figure 1c shows that the genetic gain in β of yield-ability from 1926 to 2010 was 10.2 kg/ha/year among all 237 core cultivars combined and more than double that value, at 21.8 kg/ha/year, among the 23 major cultivars. In terms of the yield-plasticity (α), the genetic gain of the 237 core cultivars was not significant, but interestingly, that of the 23 major cultivars was marginally significant ($P = 0.065$), with a 1.1% annual increase relative to the mean plasticity of the 23 cultivars (Fig. 1d). This finding suggests that local farmers have been choosing cultivars not only for higher potential yield but also with the intent to obtain higher yields to cope with a changing climate. In contrast, Japanese rice breeders have continuously improved only the potential yield of their rice cultivars.

It is worth noting that we observed no significant relationship between the potential yield and yield-plasticity of all 237 cultivars combined or of the 23 major cultivars (Fig. 1e). This result suggests that the two traits are genetically controlled independently.

**Exploring new genomic regions related to yield by means of a GWAS.** Principal-components analysis was performed to test the genetic background that might be responsible for the false positives (Supplementary Fig. S12). PC1 separated the cultivars bred in the northern region of Japan (Hokkaido) from those bred in

other regions of Japan. We performed the GWAS by using a mixed linear model with covariates to correct for population structure so as to identify genomic regions associated with yield for either α or β. A significant peak associated with α was detected on the short arm of chromosome 10 (Fig. 3a and Supplementary Figs. S12 and S13). In the GWAS for β, which had a high heritability (Fig. 2c), the GWAS did not identify a significant peak, but several possible peaks corresponding to chromosomes 9, 11, and 12 were identified (Fig. 3b and Supplementary Fig. S13). Local Manhattan plots and linkage disequilibrium (LD) analysis of the significant peak associated with α showed that the genotypes of the five significant SNPs were highly correlated, suggesting that the candidate region was within 3–4 Mb on chromosome 10 (Fig. 3c). We classified the 237 core cultivars into five haplotypes based on the genotype of five significant SNPs, excluding cultivars with heterogeneous genotypes (Fig. 3d). The cultivars carrying Hap5 showed a higher α value than cultivars carrying Hap1 and Hap2 (Fig. 3e and Supplementary Data 4).

## Discussion

We successfully standardized the cultivar characteristics of rice yield by using $Y_P$ from the CGM to account for environmental effects on the observed yield data, and then used these data to characterize two genotypic coefficients that describe yield: α and β. We call this the YpCGM method. This method extends the pioneering FW method[11] and allows the use of independent yield data measured at different sites and in different years as inputs for CGM analysis. In fact, the two parameters quantified by using a 'big data' approach were then used for genomic prediction (Fig. 2) and for exploring new genomic regions related to yield by means of GWAS analysis (Fig. 3). Our YpCGM method allowed the use

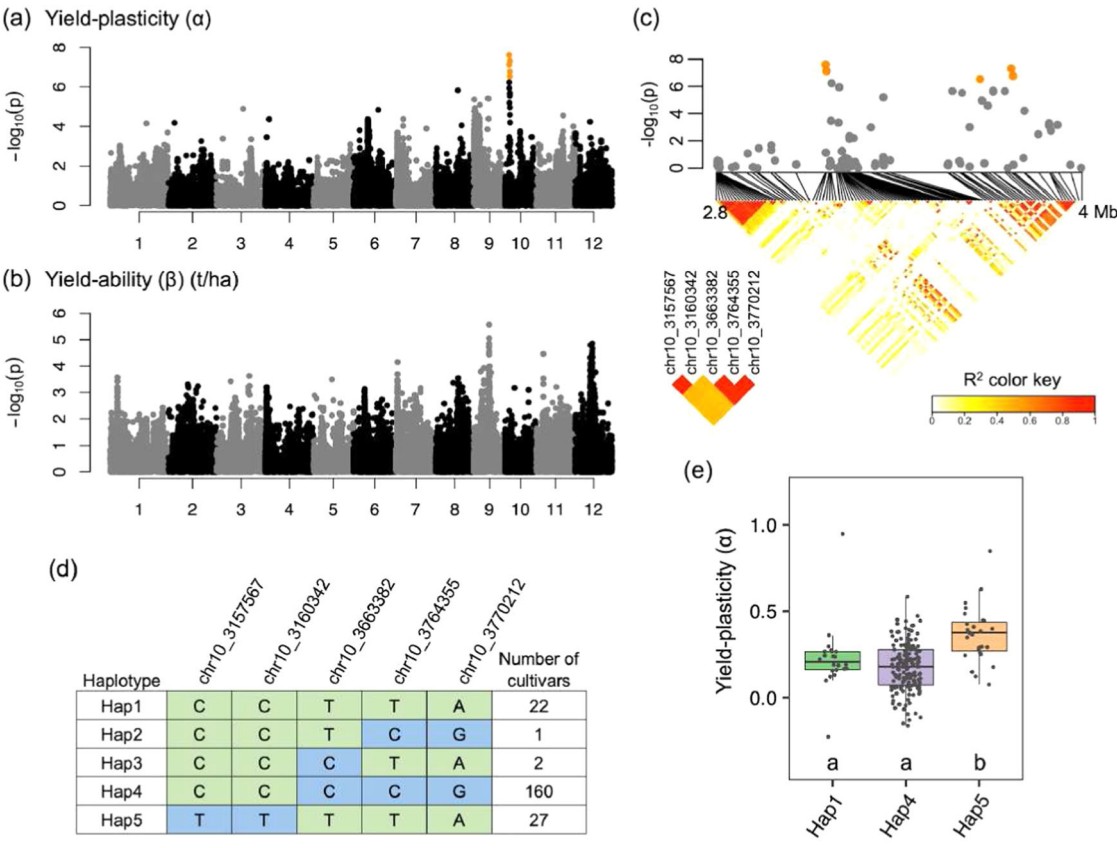

**Fig. 3 The genome-wide association study for yield-plasticity and yield-ability. a** Manhattan plot for yield-plasticity. Orange dots indicate significant SNPs. A genome-wide significant threshold –log10 (p) = 6.191 was determined by the Bonferroni multiple test correction $q < 0.05$. **b** Manhattan plot for yield-ability. The Manhattan plots were visualized with the R gaston package[68]. **c** Local Manhattan plot (upper) and LD heatmap (lower) surrounding the five significant SNPs (orange dots in the local Manhattan plot) on chromosome 10 ranging from 2.8 to 4 Mb for yield-plasticity. The LD heatmap on the lower left indicates the correlation among the five significant SNPs. The white to red gradient (lower panel) indicates the range of $R^2$ values. **d** DNA polymorphism of the five significant SNPs and haplotypes (Hap) based on them. **e** Boxplots for yield-plasticity based on the haplotypes. Hap2 and Hap3 had a small number of cultivars and were excluded. Box edges represent the 0.25 and 0.75 quantiles, with the median values shown by bold lines. Whiskers extend to data no more than 1.5 times the interquartile range. Differences between the haplotypes were analyzed by the Steel–Dwass test, and the same letter indicates no significant difference at $\alpha = 0.05$.

of a large volume of yield data, and allowed new insights obtained by means of upcycling yield data from a large number of trials.

Our GWAS let us explore genomic regions related to yield, and the results can provide guidance for future breeding targets. However, one limitation of GWAS is that the analyses require a large quantity of yield phenotype data for a large number of cultivars (e.g., ref. [25]). Our new method may let breeders use the large number of available historical records to calculate the two genotypic coefficients that we identified. Figure 3 shows the results of our GWAS and the ability of this method to identify new genomic regions associated with yield. The significant peaks detected for α justify additional research using our approach. Although we did not detect significant peaks for β, our analysis suggested the existence of several non-significant peaks that deserve additional study. One site on chromosome 1 for potential yield (position 5,484,598; $-\log_{10}(p) = 3.72$) is close to the yield-related gene *Gn1a*[26] for cytokinin oxidase regulation and panicle morphology. In addition, peaks on chromosome 2 (position 15,039 574; $-\log_{10}(p) = 4.02$) and chromosome 12 (position 12 792 478; $-\log_{10}(p) = 4.37$) are close to the respective locations of *TAC4*, which controls tiller angle by regulating the endogenous auxin content[27], and *T20*, which encodes ζ-carotene isomerase and is related to tiller formation[28]. Four additional previously unknown regions for potential yield may exist. Our methodology therefore appears to be a useful way to do GWAS supported by

other forms of genomic analysis such as detection of quantitative trait loci by upcycling data from previous research, without requiring slow and labor-intensive field trials.

Genetic gain analysis revealed a significant improvement in β in response to breeding efforts (10.2 kg/ha/year among all 237 core cultivars) and even greater gains from local farmers' cultivar choices (21.8 kg/ha/year among the 23 major cultivars) (Fig. 1c). The magnitude of the improvement was similar to the range reported by ref. [29]. in Hokkaido, at 21 to 29 kg/ha/year among the eight cultivars that were introduced between 1905 and 1988 in a 2-year trial, and the results of Zhang and Kokubun[30], at 17 kg/ha/year among 10 cultivars introduced between 1893 and 1991 at three sites with different environments in the Tohoku region (calculated from their Fig. 1).

In terms of yield-plasticity (α), genetic gain was apparent among the 23 major cultivars, with a 1.1% annual increase relative to the mean plasticity, but not among the 237 core cultivars (Fig. 1d). These results suggest that local farmers have been choosing cultivars not only for higher potential yields, but also with the goal of obtaining higher yields despite a changing climate. A similar breeding direction to increase yield-plasticity in wheat cultivars was reported in Argentina, Australia, Italy, the UK[31], and the USA[32]. We hypothesized that a cultivar with higher yield-plasticity may increase yield when grown in Japan with no constraints imposed by water availability but with constraints from climate change. We

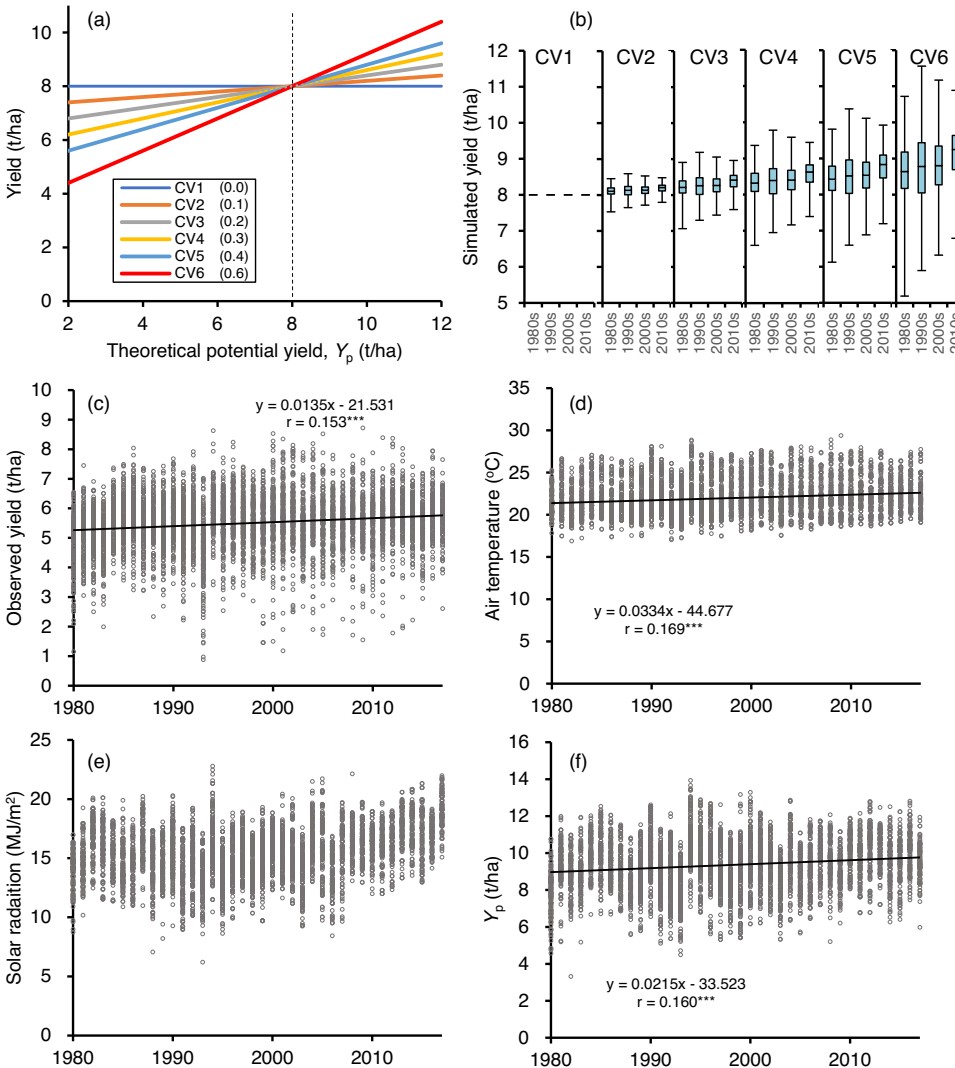

**Fig. 4 Simulations of the role of phenotypic plasticity in yield under different environments based on field trial data from the 1980s to the 2010s in Japan. a** Scenarios for cultivars tested with six yield-plasticity ($\alpha$) levels (numbers in parentheses) at fixed yield-abilities ($\beta$) of 8 t/ha in response to the theoretical potential yield ($Y_p$) estimated by the weather-driven crop growth model. **b** Simulated average yields in the six plasticity scenarios for four decades (1980s, $n = 1631$; 1990s, $n = 1957$; 2000s, $n = 1629$; 2010s, $n = 1125$). **c** The observed yield of 'Koshihikari' ($n = 6342$). **d** Observed mean air temperature before heading during the simulation period. (**e**) Observed mean solar radiation before heading. **f** $Y_p$ for the 6342 environments that 'Koshihikari' experienced during the simulation period. ***$P < 0.001$.

tested six scenarios based on different yield plasticities ($\alpha$), with values ranging from 0.0 to 0.6, in scenarios CV1 to CV6 (i.e., six $\alpha$ values within the observed range) (Fig. 4a). These results are based on the assumption of a fixed yield-ability (here, $\beta = 8$ t/ha) in 6342 field trial environments using the cultivar 'Koshihikari' and data from 1980 to 2017. Simulation CV1, with zero plasticity, provided a constant yield of 8 t/ha; increasing the plasticity increased the yield, with the maximum increase in scenario CV6, with $\alpha = 0.6$ (Fig. 4b and Supplementary Data 5). The simulated yield increased with increasing plasticity throughout the simulation period, with the greatest gain in the 2010s, at 14%, versus the 1980s, at 8%. This simulation was based on historical weather records and suggested that yield-plasticity is a promising future breeding target with which to accelerate genetic gain in rice yield as an adaptation to climate change in Japan. Similar positive effects of higher yield-plasticity were observed in scenarios in which $\beta$ was set to 6 to 9 t/ha in increments of 1 t/ha.

In fact, environmental conditions in Japan improved for 'Koshihikari' from 1980 to 2017, leading to an observed yield increase of 13.5 kg/ha/year ($n = 6342$; Fig. 4c). This trend may have resulted from an increase in the mean air temperature before heading, of 0.03 °C/year in the range from ca. 18 to 26 °C (Fig. 4d), but without a significant trend in solar radiation (Fig. 4e). $Y_p$, which reflects the cumulative effect of environmental resources, increased by 21.5 kg/ha/year (Fig. 4f) even without accounting for the $CO_2$ fertilization effect in this model[33–35].

This simulation was based on historical weather records and suggested that yield-plasticity is a promising future breeding target for accelerating genetic gain in rice yield as an adaptation to climate change in Japan since plant growth under favorable environmental conditions generally benefits from increased availability of resources, including temperature[36,37], atmospheric $CO_2$[33–35,38], and solar radiation[39]. Jagermeyr et al.[40] predicted increased rice yields globally from 2069 to 2099 under future climate scenario SSP585, especially at higher latitudes but with a decreased yield at lower latitudes. Their prediction means that environmental resources will become more suitable for rice at higher latitudes. The genetic control of yield-plasticity should be a

powerful option to compensate for the predicted yield losses at lower latitudes and could boost global crop productivity under future climatic conditions if this plasticity can strengthen abiotic stress tolerance[41]. This approach, combined with stronger tolerance to abiotic stresses to mitigate the effects of unusual climate events, which are expected to increase in frequency and severity, could play a key role in securing global food security for the predicted world population of 9 billion people in 2050.

Compared to previous approaches using CGM to evaluate cultivar effects on crop yield, the YpCGM method has certain limitations and advantages. A CGM can predict crop yield by integrating mathematical descriptions of plant physiological processes in response to changes in their environment and in management practices throughout the cropping cycle. In general modeling studies, cultivar characteristics are expressed by the cultivar-specific parameters used in the mathematical equations that describe a plant's physiological processes, and the parameters are used for genomic analysis of plant yield[19,42,43], leaf expansion[44], and flowering time[45,46]. Our method eliminates the need to parameterize the many process models in a CGM by combining easily available climate data with parameterization of two cultivar-specific variables, $\alpha$ and $\beta$, that determine the overall ability of yield formation by means of linear regression in response to $Y_p$ calculated with fixed default parameters in the CGM for the attainable yield to represent the effects of the growing environment.

One disadvantage with our approach is that, unlike previous modeling studies that required parameterization of coefficients for physiological processes, it cannot identify the causal physiological processes that explain differences among cultivars in general modeling studies. Also, the relatively low heritability of $\alpha$ (Fig. 2c) might have resulted from the cumulative error in each physiological process caused by pooling many studies. This drawback is offset by the ability of our method (a 'big data' approach) to compensate for limited data from studies that examined many cultivars simultaneously across a range of environments to parameterize the CGM's process models.

Recently, Jighly et al.[47]. examined a method that partially parameterized cultivar-specific parameters in a CGM. They measured five traits (grain size, grain number, grain protein content, phenology, and yield) of 590 wheat cultivars in six field trials from 2017 to 2020 in Australia. From this parameterization, they estimated cultivar-specific parameters in the ABIOGP-Wheat model and successfully used this approach for genomic prediction. Kadam et al.[48]. attempted a similar analysis in rice for five traits (plant height, grain size, ripening percentage, yield, and phenology) of 267 cultivars using the GECROS model. They used default parameters in the model for leaf area development, radiation use efficiency et al. except for parameters that reflected differences between cultivars. The method of partially parameterizing a CGM has the advantage of revealing the causal factors for cultivar-specific yield variation. However, in most of 'big data', the data required to accurately parameterize a model may not be available. Our new method can provide sufficiently good data to accelerate the breeding efficiency to produce cultivars adapted to future climates.

The YpCGM method can be improved. First, we used only a single CGM (Supplementary Fig. S5)[49] to calculate $Y_p$. This means that $\alpha$ and $\beta$ include an error component caused by the structure of the model, which, for example, does not account for soil type and fertility[50], atmospheric $CO_2$[20], or the temperature of the irrigation water[51]. Combining $Y_p$ from multiple models might improve the prediction accuracy. Second, future research should test the potential of ensemble multi-model analysis using models with different complexities and abilities to account for specific physiological

processes, thereby increasing the accuracy of the $Y_p$ calculation, as was done recently in simulations of future crop yield[40,52–54]. Third, we used observed phenology data as input data to improve the accuracy of parameterization for cultivar-specific variables. The incorporation of phenology sub-models would let us use additional data when phenology data is not available. This could also extend our methods to test the performance of cultivars in environments where they have not previously been tested and to predict their performance under future climates. Fourth, our new approach is made possible by the sacrificed statistical power of $n$ in the big data, which compiled data on the yield variations in 72,510 trials of a large number of cultivars, then performed linear regression for 237 cultivars to identify the values of the two genotypic coefficients. Using the full dataset of 72,510 individual yield trials would improve the model's ability to account for G × E interactions and the genotypic characteristics of each cultivar. Recently, several authors tried to improve the accuracy of GP for genomic and phenotyping data by using intermediate secondary traits of means of environmental parameters during a specific growth period or to improve a model's output[2,6,8,9]. In future research, it may be possible to extend our approach to extract cultivar-specific values of $\alpha$ and $\beta$ by using data from all 72,510 yield trials of the 237 cultivars.

The YpCGM method may be applicable to other crop species. We have shown the validity of this method for rice in Japan grown under no water constraints and relatively uniform soil and management conditions. This captured the variation of yields, and let us use a simple empirical model that requires daily solar radiation and air temperature as inputs (Supplementary Fig. S5). To apply the YpCGM method more widely, it's necessary to use a CGM capable of expressing the performance of different crop species under diverse environmental conditions. Many CGMs have been developed for specific species such as wheat, soybean, maize, and sorghum. These include APSIM, the Agricultural Production Systems sIMulator[55], and DSSAT, the Decision Support System for Agrotechnology Transfer[56], and species-specific crop models for wheat[54], maize[52], and rice[53]. Large phenotype datasets are available for crop species such as wheat[3,17], maize[2,5], sorghum[10], chickpea[7], and common bean[4]. These data are based on field trials conducted under a diverse set of growing conditions. These datasets can be analysed by using the YpCGM method to upcycle previous phenotyping data and increase breeding efficiency by calculating the genotypic coefficients for yield-ability and yield-plasticity.

We developed a new method for mitigating the effects of the genotype × environment (G × E) interaction by developing the YpCGM method. In this approach, we used a weather-driven crop growth model to combine data from independent yield trials conducted under varying environmental conditions, with the model's estimated potential yield based on weather data. This approach let us reanalyse a large quantity of valuable data from previous trials. To the best of our knowledge, we are the first to use this approach to mitigate the effects of the G × E interaction and characterize the productivity of cultivars by integrating data from many trials across a range of environments using only two regression coefficients. The YpCGM method upcycles accumulated yield data from measurements such as genetic gain analysis and GWAS to guide future breeding programs toward developing new cultivars suitable for the world's changing climate. Because our analysis is only valid for a specific set of conditions (i.e., flooded rice in temperate Japan), the model should not be used to extrapolate beyond the conditions included in the range of field trials we studied. However, the approach could, in principle, be repeated for other combinations of conditions to provide new values of the two parameters.

## Materials and methods

**Phenotype data.** We obtained yield datasets for rice (*Oryza sativa* L.) from 207,331 trials with 8524 cultivars during the 38 years from 1980 to 2017. The data were obtained from field trials at 110 public agricultural experimental stations in Japan conducted by the Institute of Crop Science of the National Agriculture and Food Research Organization, Japan (NARO; 2017 version)[57]. From the database, we selected 237 core cultivars, for a total of 72,510 yield datasets, as follows: (i) From the pedigree network information of 14,032 rice cultivars, we selected 200 cultivars that were central nodes in the classification by the *k*-medoids method (*k* = 200) (Supplementary Fig. S3); (ii) From the hierarchical clustering of the pedigree relationship matrix using the Wards method in R, we selected 158 cultivars with as many phenotypic records as possible, one cultivar from each cluster; (iii) Finally, from this total of 358 cultivars, we selected 237 core cultivars for which data were available from more than 20 trials per cultivar in a location × year combination (to maintain the accuracy of the regression analysis when the number of trials was incrementally increased) and for which seed was available at the gene bank. Supplementary Fig. S2 shows the locations of the study sites, and Supplementary Data 6 provides geographical details, time periods, and numbers of cultivars. The data availability for each of the 237 cultivars ranged from 20 to 6342 datasets (with a median of 163 and an average of 306).

We cleaned the phenotypic data to remove outliers by excluding values that lay more than 4.0 standard deviations from the median value for each cultivar. The 237 cultivars included the oldest released ('Asahimochi', 1926) and the most recently released ('Tachiharuka' and 'Yukigozen', 2010) (Supplementary Data 7 and 8). Each cultivar provided data from 20 to 6342 yield trials that reported phenotype datasets (yield, days to heading and maturity, panicle number, and panicle length). Rice plants were grown in paddies under flooded conditions and with consistent management practices; each study used the best management practices at the time for site preparation, fertilization rate, and pest and disease control. Because the cultivation methods were similar between studies, but not identical, future research should determine whether these differences may have decreased the accuracy of our analysis. All yields were converted to a 14% moisture content.

**YpCGM method.** In the YpCGM method, we standardized the yields of each cultivar by using the theoretical potential yield ($Y_p$). We used a weather-driven crop growth model (Supplementary Fig. S1b). The standardized values are calculated using the slope and estimated yield at the *SPY* (t/ha) to quantify the cultivar-specific characterization of yield using two parameters: the yield-plasticity ($\alpha$; dimensionless) and the yield-ability ($\beta$; t/ha) from previous field trials:

$$
\begin{aligned}
Y_{\mathrm{obs}(i,j)} &= \alpha_{(i)} Y_{p(i,j)} + b_{(i)} \\
&= \alpha_{(i)}(Y_{p(i,j)} - SPY) + \beta_{(i)}
\end{aligned}
\tag{1}
$$

where $Y_{\mathrm{obs}(i,j)}$ and $Y_{p(i,j)}$ represent the observed and potential yield, respectively, of cultivar $i$ in environment $j$ (year, location, and management regime), and $b_{(i)}$ is the intercept at $Y_p = 0$. Our conversion of equation 1 to a form that uses *SPY* lets us characterize the yield-ability $\beta_{(i)}$, which represents the *x*-axis intercept at $Y_p = SPY$; thus, it is the expected yield at $Y_p = SPY$. *SPY* can be set by accounting for the majority (more than 85%) of the $Y_p$ experienced for the tested cultivars. This result covers the observed range of yields and provides a straightforward understanding of the productivity of a given cultivar. We set *SPY* to 8 t/ha because 85% of all cultivars (201 of the 237 cultivars) had experienced $Y_p = 8$ t/ha (Supplementary Fig. S4). The value of $Y_p$ among the 237 cultivars averaged 5.0 t/ha, with a minimum of 0.1 t/ha and a maximum of 9.8 t/ha. We called $\alpha_{(i)}$, $\beta_{(i)}$ and $b_{(i)}$ as genotypic coefficients in the followings.

We estimated $Y_p$ for a cultivar in each year and at each location by using a simple empirical CGM developed by Masuya and Shimono[49] (Supplementary Fig. S5). The model integrates daily canopy radiation capture and radiation use as a function of the daily air temperature and solar radiation[58]. Leaf senescence is expressed as a function of photosynthate partitioning, radiation-use efficiency, and spikelet fertility[51]. In the cold stress sub-model, the cooling degree-days for inducing spikelet sterility (at a base temperature of 20 °C) was calculated during the reproductive growth phase[59]. In the development sub-model, we predicted the phenology of a developmental index ($DVI$[60]); and combined this model with the observed heading and maturity dates. The $DVI$ on each date was calculated from the air temperature by using the ratio of the actual temperature to the cumulative effective air temperature (here, defined as >10 °C) for each period between the observed transplanting, heading, and maturity dates. This calculation eliminated the need for daylength input data and cultivar-specific parameters that accounted for different phenological responses. This procedure also let us standardize the yield per unit area by using the observed solar energy capture even for identical cultivars grown in different years and at different locations and for cultivars with different maturity dates. Note that other abiotic stresses (heat, drought, and atmospheric $CO_2$) that influence rice yields in Japan were not considered in our model to simplify the calculations. Also, the effects of soil fertility and fertilization were not considered, because all agricultural experimental stations grew rice within the range of optimal nutrient conditions. The daily air temperature and solar radiation during the study periods at the weather station closest to each of the 110 locations that provided yield data for 38 years were obtained from the MeteoCrop database (https://meteocrop.dc.affrc.go.jp/real/top.php) (Supplementary Fig. S2).

The accuracy of the predictions of observed yield by $Y_p$ was evaluated with the root-mean-square error (*RMSE*) for each cultivar compared to the cumulative air temperature and solar radiation throughout the season, and the observed panicle number and length.

To validate the robustness of the regression model (1), the regression parameters $\alpha_{(i)}$ and $\beta_{(i)}$ were estimated for each cultivar $i$, excluding data from environment $j$, and Y in environment $j$ was predicted based on the estimated parameters. That is, the prediction ability of the regression model (1) was evaluated via leave-one-out cross-validation. To evaluate the accuracy of the predictions, we calculated the coefficient of determination in fitting (the ratio of the variation in the fitted value of Y to the total variation) and the coefficient of determination in prediction (the ratio of the variation in the predicted value of Y to the total variation) and compared the coefficient values. If the difference between them is small, the regression model (1) is expected to make robust predictions without overfitting.

**Genotype data.** We resequenced 166 rice cultivars (Supplementary Data 9). First, we extracted genomic DNA from young leaf tissue of each cultivar with NucleoSpin Plant II kits (Macherey-Nagel GmbH & Co. KG, Düren, Germany). We then quantified the DNA with a Qubit fluorometer (Invitrogen, Waltham, MA, USA). Next, we constructed libraries with Riptide High Throughput Rapid DNA Library Prep kits (iGenomX, Carlsbad, CA, USA), following the manufacturer's protocol. We sequenced the Riptide libraries on the Illumina NovaSeq platform (Illumina, San Diego, CA, USA), using 150-bp paired-end reads, and then demultiplexed the results in fgbio v. 1.3.0 software (https://github.com/fulcrumgenomics/fgbio). We sequenced additional cultivars with a low number (<6,300,000) of sequence reads on the Illumina HiSeq platform. In addition, we obtained FASTQ files from the DNA Data Bank of Japan Sequence Read Archive (DRA) for 70 rice cultivars[61,62].

The raw sequence reads of 236 cultivars were cleaned to improve their quality (quality trimming and adapter clipping) using the Trimmomatic software v. 0.39[63] with the options 'PE -phred 33 ILLUMINACLIP:TruSeq3-PE.fa:2:30:10 LEADING:20 TRAILING:20 SLIDINGWINDOW:4:15 MINLEN:75'. We retained only paired output reads. After these pre-processing steps, we mapped the remaining reads onto the 'Nipponbare' reference genome IRGSP-1.0, using the *bwa mem* command in BWA[64], with the options '-a -T 0'. We obtained coordinate-sorted files in BAM format using the SAMtools *sort* command in SAMtools v. 1.9[65]. We obtained BAM files that contained only correctly oriented and properly paired mapped reads by filtering the specified bit in the FLAG field during scanning with the SAMtools *view* command. The BAM files were filled in as mate coordinates using the SAMtools *fixmate* command. Finally, we obtained BAM files that contained only paired mapped reads by using the samtools *view* command again.

SNP-based genotype calling can be obtained as a file in variant call format (VCF). First, the VCF file was generated as follows: (i) BCFtools v. 1.9[65] mpileup command with the options '-a DP,AD,SP,ADF,ADR -B -q 10 -Q 13 -C 50'; (ii) BCFtools call command with the option '-vm -f GQ,GP'; and (iii) BCFtools filter command with the option "-i "INFO/MQ > = 10"". Next, we only retained reliable SNPs where (1) the SNP had a frequency of 0.8 or greater for samples with a depth of 5 or higher and a genotype quality of 20 or higher, and (2) the mapping quality was 30 or higher. Subsequently, we converted the genotypes with low depth and low genotype quality to 'missing' using VCFtools[66] with the options '--minGQ 20 --minDP 2'. Afterward, we entered the resulting genotype dataset in BEAGLE v. 5.1[67]. Furthermore, we filtered out SNPs where the minor allele frequency (MAF) was lower than 0.05 among the 237 samples. Finally, we completed a core cultivar genotype dataset with 91,800 SNPs from 237 cultivars, including 'Nipponbare'.

**GP analysis: prediction of genotypic coefficients and calculation of heritability.** The accuracy of the predictions based on genomic and pedigree data for the genotypic coefficients $\alpha$ and $\beta$ was evaluated based on the genomic (realization) relationship matrix and the pedigree (numerator) relationship matrix. We tested the genomic best linear unbiased prediction (gBLUP), the pedigree-based best linear unbiased prediction (pBLUP), a combination of the two approaches (gpBLUP), and a prediction based on the genome × pedigree information (g×pBLUP). In all BLUP modeling, we built a multiple trait model that allowed us to account for correlations between $\alpha$ and $\beta$.

The accuracy of the predictions based on genomic and pedigree data for the genotypic coefficients $\alpha$ and $\beta$ was evaluated. Predictions were based on the genomic (realization) relationship matrix and the pedigree (numerator) relationship matrix.

The genomic relationship matrix was calculated with the same set of SNPs used for the GWAS (91,800 SNPs satisfying the condition $MAF \geq 0.05$). Specifically, the value was calculated as $\mathbf{G} = \mathbf{XX'}/m$ from the matrix $\mathbf{X}$ representing the SNP genotypes of the 237 cultivars, where the $(i, j)$ elements of $\mathbf{X}$ represent the genotype of the $j$-th SNP of the $i$-th cultivar, first scored as 0 for reference-type homozygous, 2 for non-reference-type homozygous, and 1 for heterozygous; they were then standardized to have a zero mean and unit variance, and where $m$ represents the number of SNPs. The pedigree relationship matrix was calculated based on the parental information of each cultivar obtained from the NARO (https://ineweb.narcc.affrc.go.jp/) rice cultivar database. The pedigree relationship matrix, $\mathbf{A}$, was

calculated based on pedigree information of 15,145 cultivars, including lines that were included as dummies to account for backcrossing steps.

Prediction models were constructed using the genomic and pedigree relationship matrices. Specifically, the following four BLUP models were constructed as shown below:

gBLUP:

$$\mathbf{y} = \boldsymbol{\mu} + \mathbf{g}_g + \boldsymbol{\varepsilon} \tag{2}$$

pBLUP

$$\mathbf{y} = \boldsymbol{\mu} + \mathbf{g}_p + \boldsymbol{\varepsilon} \tag{3}$$

gpBLUP

$$\mathbf{y} = \boldsymbol{\mu} + \mathbf{g}_g + \mathbf{g}_p + \boldsymbol{\varepsilon} \tag{4}$$

g×pBLUP

$$\mathbf{y} = \boldsymbol{\mu} + \mathbf{g}_g + \mathbf{g}_p + \mathbf{g}_{gxp} + \boldsymbol{\varepsilon} \tag{5}$$

where $\mathbf{y}$ is the vector with a length of $n \times 2$ ($n$ genotypes and two YpCGM parameters) whose $i$-th and $n + i$-th elements are $\alpha_{(i)}$ are $\beta_{(i)}$ (or $\alpha_{(i)}$ or $b_{(i)}$), respectively, $\boldsymbol{\mu}$ is the vector whose elements are the overall mean, $\mathbf{g}_g$ is the vector of genetic effects accounted for by the between-parameter covariance and the genomic-based relationships, i.e., $\mathbf{g}_g \sim N(\mathbf{0}, \boldsymbol{\Sigma} \otimes \mathbf{G}\sigma_G^2)$, where $\boldsymbol{\Sigma}_G$ is the genomic-based genetic covariance matrix of the two parameters and $\otimes$ is the kronecker product, $\mathbf{g}_p$ is the vector of genetic effects accounted for by the pedigree-based relationships, i.e., $\mathbf{g}_p \sim N(\mathbf{0}, \boldsymbol{\Sigma}_A \otimes \mathbf{A}\sigma_A^2)$, where $\boldsymbol{\Sigma}_A$ is the pedigree-based genetic covariance matrix, and $\mathbf{g}_{g \times p}$ is the vector of genetic effects accounted for by the interaction between genomic- and pedigree-based relationships, i.e., $\boldsymbol{\gamma} \sim N(\mathbf{0}, \boldsymbol{\Sigma}_{GA} \otimes (\mathbf{G} \odot \mathbf{A}))$, where $\boldsymbol{\Sigma}_{GA}$ is the genetic covariance matrix of the interaction effects and $\odot$ is the element-wise product (the Hadamard product) of the matrices, and $\boldsymbol{\epsilon}$ is the vector of residuals assuming $\boldsymbol{\epsilon} \sim N(\mathbf{0}, \mathbf{R} \otimes \mathbf{I})$, where R is the residual covariance matrix between two coefficients. The g×pBLUP is the full model with interaction terms between genomic and pedigree relationships (Howard et al., 2019). The *MTM* function in the R package *MTM* (de los Campos and Grüneberg 2016) was used to estimate the parameters of the model and to calculate BLUP values. In the estimation, a total of 20,000 iterations of a Gibbs sampler were run, and the first 5000 iterations were discarded.

To evaluate the model's prediction accuracy, we conducted 10 replicates of 10-fold cross-validation. Specifically, we estimated the parameters of each prediction model, calculated BLUPs for cultivars in 9 of the folds, and then predicted the $y$ value for cultivars in the one fold that had been left out. This step was repeated for all folds. Predicted values for all 10 folds were aggregated, and correlations between observed and predicted values and RMSE values were calculated. This procedure was repeated 10 times.

To evaluate the levels of genetic control of the genotypic coefficients α and β, we calculated heritability using genomic and pedigree relationship matrices. One estimate was obtained for each cultivar for each of the genotypic coefficients, and there were no replications. Thus, heritability was calculated using the genetic variance (a diagonal element of $\boldsymbol{\Sigma}_G$ or $\boldsymbol{\Sigma}_P$) and the residual variance (a diagonal element of $\mathbf{R}$) estimated for the gBLUP and pBLUP models in each genetic coefficient. Specifically, when $\sigma_G^2$ was the diagonal element of $\boldsymbol{\Sigma}_G$ or $\boldsymbol{\Sigma}_P$ corresponding to the parameter of interest (α, β or b) and $\sigma_G^2$ was the diagonal element of $\mathbf{R}$ corresponding to the parameter, the ratio $\sigma_G^2/(\sigma_G^2 + \sigma_E^2)$ was calculated as an estimate of the heritability of the parameter. In the calculation, the parameters of the gBLUP and pBLUP models were estimated using data from all 237 cultivars. The *MTM* function in the *MTM* R package (de los Campos and Grüneberg 2016) was used for the calculation. The iterations in the Gibbs sampling were the same as the iterations used for calculating the BLUP parameters.

The accuracy of the predictions based on genomic and pedigree data for the genotypic coefficients α and β was evaluated based on the genomic (realization) relationship matrix and the pedigree (numerator) relationship matrices.

**GWAS analysis**. We performed a GWAS of the α and β of the 237 cultivars listed in Supplementary Data 7 using the genotype set of 91,800 SNPs. Population structure was estimated (i.e., calculation of PCA [$n = 4$] and kinship [K] matrixes) and a GWAS based on the mixed linear model was performed in the R package *gaston*[68]. A genome-wide significance threshold, $-\log_{10}(p) = 6.191$, was determined by a Bonferroni-adjusted multiple test correction ($q < 0.05$). The R *gaston* package visualized the Manhattan, Q–Q, and local Manhattan plots. LD-heatmaps were generated in the R *LDheatmap* package[69]. Statistical analysis to compare the haplo-types for α was performed in Python v. 3.10.4. The Shapiro–Wilk test (the *shapiro* function from the stats library from the *scipy* package) was used to check that the data were normally distributed. The Levene test (the *levene* function from the stats library of the *scipy* package) was used to confirm the homogeneity of variance. The data were not normally distributed, nor was the variance homogeneous. Means were therefore compared by Kruskal–Wallis test (the *kruskal* function from the *stats* library of the *scipy* package) followed by the Steel–Dwass test (the *posthoc_dscf* function from the *scikit_posthocs* package). All statistical significances were set to $P < 0.05$.

**Statistics and reproducibility**. Graphs were generated using Microsoft Excel 365 (Figs. 1, 2, 4) or R v4.2.1 (Fig. 3).

All graphs show the mean as a bar or fitted curve and individual data points as a scatter plot.

The analysis used a value for each of 237 cultivars obtained from regressions of 20–6342 field trials across locations and years. The source data for all graphs presented are included as Supplementary Data.

**Reporting summary**. Further information on research design is available in the Nature Portfolio Reporting Summary linked to this article.

## Data availability

The sequence reads for SNP genotyping used in this study were deposited at the DNA Databank of Japan (DDBJ, https://www.ddbj.nig.ac.jp/index-e.html) (Supplementary Data 9). The SNP genotype dataset from the 237 rice cultivars used in this study is available at https://doi.org/10.5281/zenodo.7593964. The source data for the graphs that are presented are in Supplementary Data. All other data are available from the corresponding author on reasonable request.

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

## Acknowledgements

We thank Emeritus Prof. M. Okada of Iwate University for his helpful suggestions. We also thank M. Ozaki, A. Oyamada, Dr. Y. Masuya, K. Shioi and K. Nagai, Iwate University. We also thank K. Ito, Y. Ogasawara and S. Natsume, Iwate Biotechnology Research Center, for their technical support. We also thank the following organizations for providing seeds of the rice cultivars: the Genebank at the National Agriculture and Food Research Organization, Japan (NARO), NARO rice breeding sections, and local public agricultural research stations at the Hokkaido, Aomori, Iwate, Yamagata, Tochigi, Chiba, Nagano, Niigata, Toyama, Fukui, Shiga, Aichi, Hyogo, Miyazaki, and Kagoshima prefectures In this research project, we used the NIG supercomputer at the ROIS National Institute of Genetics and the supercomputer of ACCMS, Kyoto University. This work was supported by a Japan Society for the Promotion of Science KAKENHI Grant (Number JP 19H00938).

## Author contributions

H.S. conceived the idea for the study. H.S., C.S., A.A., C.H.K. and H.I. analysed the data and participated in the interpretation of the results and manuscript preparation.

## Competing interests

The authors declare no competing interests.

## Ethics approval

The authors declare that all methods were performed in accordance with the relevant guidelines and regulations. The following organizations provided the test materials: NARO and local public agricultural research stations at Hokkaido, Aomori, Iwate, Yamagata, Tochigi, Chiba, Nagano, Niigata, Toyama, Fukui, Shiga, Aichi, Hyogo, Miyazaki, and Kagoshima prefectures. We obtained permission from all organizations to use these cultivars.
