## [Peer Review File · Communications Biology]

Reviewers' comments:

Reviewer #1 (Remarks to the Author):

The author used a large historical rice data within a new simple crop growth model named YpCGM that involves only two critical genotype-specific parameters. The authors then applied genomic prediction and GWAS on the simulated parameters but their prediction accuracies were considerably low. I have a mixed feeling about this paper and I cannot say that I can accept it or rejected it before getting clarification from the authors on multiple issues stated below. Specifically, the description of the model is not very clear and it may not be practical to predict the performance of cultivars in unobserved environments as it requires some measured phenotypes as inputs. Moreover, based on what described in equation one, it may suffer from parameter equifinality.

Please find the below comments:

Generally, the paper need to be fully restructured. Each section should contain what it supposed to include, e.g., introduction doesn't need to have a description of the method.

Introduction:

In general, the introduction is very long and some parts of it can be moved to the materials and methods (e.g. around the equation) or the results/discussion (the following paragraphs).

Line 104: CGM-WGP model can be applied to big data with limited number of phenotyped traits as well as limited number of phenotypic records per cultivar (see Jighly et al. 2022 <https://doi.org/10.1093/jxb/erac393>).

Materials and methods:

It is not clear if YpCGM uses yield-ability and plasticity a single type only in equation 1 or they are used in multiple equations along the whole model. If they are only using both of them in a single equation, then you will have a parameter equifinality problem

Line 196-197: Did you considered the observed heading and maturity directly in the model or they were used to calibrate it? If the model assumes that heading and maturity should exist, then how can you predict the performance of the cultivar in unobserved environments (which is the most important application of crop growth models)? Another problem associate with that, if you are using the observed heading and maturity, these measures are usually correlated with yield. Therefore, it will be expected to get higher accuracy for yield prediction than other "more practical" models

Line 201: If you are using the observed heading and maturity then of course the phenological parameters will not be required

Line 209: I may misunderstand something here but if you have the measured SPY for all cultivars, why don't you use it instead of setting it to a single value of 8 t/ha

Line 230: should be "< 0.5"

Line 247-250: g and p are generally highly correlated, so it will not be really useful to use them together.

Line 264&265: We usually use the term "iteration" instead of cycles or samples

Line 296: any specific python library or command?

Results

You need to present the correlation between yield-ability and plasticity with the observed phenotypes.

Line 305-309: Long sentence, please split

Line 332: I would love to see how pBLUP and gBLUP will perform if applied directly to the observed

phenotypes, e.g. yield.

Line 346-349: It might be better to describe the heritability then prediction accuracy

Line 374: Large proportion of this heading can be moved to the discussion

Discussion

Large proportion of the discussion is just a repetition of what stated in other sections

Reviewer #2 (Remarks to the Author):

GENERAL COMMENT:

This study proposes a methodology based on the integration of an empirical crop model and linear regression to better characterize rice cultivars in terms of yield potential and yield plasticity. The methodology is of interest, since it allows to use past yield and phenology data from multi-environmental trials to extract more information about GxE interactions for multiple cultivars. The dataset at the basis of the study is impressive, which enables the use of this empirical approach. The study is focused in Japan.

The manuscript is well written and clear, although some work is needed in the introduction and in the discussion sections (see detailed comments below). The description of the methodology used is appropriate to ensure reproducibility.

That said, my main concern refers to the potential extension of the approach proposed to other crops (explicitly suggested at line 441-445) or to future conditions (which is suggested throughout the manuscript). I do not agree at all on this. As explained and mentioned many time in the detailed comments here below, the statistical approach proposed in this study has the clear advantage of requiring less efforts as compared to process-based modelling approaches (no calibration for specific sub- processes needed, like e.g., leaf expansion). However, this also determines clear limits. Since the physiological processes underlying crop yield are not understood, captured and explicitly simulated, this statistical approach cannot be extended to conditions different from those explored in the field trials used. That means nor to future climates, where climate is expected to assumes unprecedented features and patterns, nor to rice or other crops not grown under flooded conditions as in this study (where temperature and solar radiation are the only limiting factors for crop growth). The validity of this methodology to conditions where factors with non-linear effects on yield play a key role is also questionable (e.g., biotic/abiotic stressors, which have not been considered in this study, line 204-205), and need to be supported by dedicated studies.

So the study is of interest and might have clear value from a breeding perspective, but the authors should also clearly highlight the limits of the study and do not speculate about the extension of the methodology to conditions different from those explored with the field trials (future climates, other crops), since this is not supported by the results of this study at all. It can be mentioned in the conclusion as a perspective, as a hypothesis to be tested and validated with further studies.

Apologies in advance for reiterating this critical point many times in the detailed comments here below, but in my opinion this is really a key issue of the manuscript.

SPECIFIC COMMENTS:

- Title: "weather-driven crop model": weather variables are among the drive variables in all crop models. I strongly suggest to use/add "empirical " because the model used in this study has a very

strong empirical component.

- L96-98: please clarify how SPY is derived.
- Introduction needs work. It seems more like a summary of the study rather than an analysis of the state of the art, of the context in which this study is placed. Please extend the analysis of the literature available beside the specific work of FW.
- L100: I do not agree. This sentence sounds like the parameterization for different physiological processes is something not actually needed and that it can be simply overlooked by using a big-data approach. Such parameterization, focused on multiple processes (e.g., leaf expansion, phenology, photosynthesis), is fundamental to avoid equifinality (i.e., same model output [in this case yield] achieved with different parameter sets), which can easily lead to loss of robustness when the crop model is applied to independent datasets.
- L104: As above: I do not agree. Sometimes it is better to have less data but of high quality rather than "big data" with no or very little understanding of the physiological processes underlying the observed yield. The empirical, black box approach proposed in this study just needs observed yield and phenology, which for sure provides the advantage of requiring less effort as compared to the other modelling approaches mentioned (no calibration for specific processes needed). However, this also determines clear limits. Since the physiological processes underlying crop yield are not understood, captured and explicitly simulated, this statistical approach cannot be extended to conditions different from those explored in the field trials (e.g., future climates, different management contexts, different crops). I would suggest at least to the authors to discuss more this issue and clearly highlight the limits of the study in the discussion section (only advantages are mentioned). I would also better underline here the specific context of this study, with yield data coming from rice grown under optimal conditions (as explained at line 149-151) which allow to undertake a data-driven approach as that described in this study.
- L110: heading date is not part of the "cultivation schedule". It is phenological development of the crop, which should be entirely simulated, not provided as input. Cultivation schedule should refer only to sowing/transplanting, fertilization, weed and pest management, harvest events (maturity date instead is still phenological development as heading date). The fact that heading date is an input, is one of the reasons why the analysis cannot be extended to new environments (e.g., future climates).
- L113-114: these are the pros, but there are also many cons like e.g., the impossibility to extend the analysis to conditions not explored with the field trials (see also previous comment).
- L127: please delete reference to future climates. With the method described, prediction cannot be extended to conditions different from those explored while developing the linear model. To do so, process-based crop models need to be used.
- L141: 20 is the minimum number of observations for each genotype, but I would suggest provide here also the maximum one. At least reference to any supplementary material where this information can be derived (Table S1). The number of observations for each genotype is a key information to evaluate the soundness of the analysis.
- L144: please clarify what 4.0 stands for or delete it.
- L204-206: the fact that heat, drought, and changes in CO₂ concentrations are not included in the study is exactly the reason why I suggest to the authors to delete from the manuscript any reference to extending prediction/support breeding for future climates. Those are expected to be key factors under future climates.
- Line 209-211: the use of observed SPY highlights once more the strongly empirical nature of the approach used.
- Line 335: prediction accuracy is pretty low. Please add to the discussion section a paragraph to discuss this issue and make comparison with available literature.
- Line 375-388: please, clarify that you refer to the changing in climate observed in the past years (those explored with your dataset). The methodology presented in this study cannot support extending the analysis to future climates (an empirical model based on statistical relationships does not support the extension to climate conditions not explored while developing the model). For this

reason, I think that calling them “climate scenarios” can be misleading, because this term usually refers to future climate projections (SSPs×GCM combinations). If you want to keep using this term, please better explain what they are/how they have been derived.

- Line 441-445: this is speculation not supported at all by the result of this study. In this study authors focused the analysis (i) on paddy rice grown under flooded conditions and optimal management (line149-151), so without any limitations to crop growth other than temperature and solar radiation, and (ii) to a restricted area (temperate climate of Japan). According to the results of the study, it is impossible to say if this method would be valid under different conditions (e.g. rice grown in warmer climates with heat waves or water scarcity issues), let alone other crops not irrigated or irrigated but not at field capacity (as rice). So, the discussion should focus on comparing and discussing results with the literature available, not on speculating about applications of the methodology without being supported by the data presented. It can be reported as a perspective, to be tested and validated in dedicated future studies.

- Line 431-436: This manuscript propose a method based on empirical modelling and linear regression. For this reason, the “advantages” highlighted by the authors (line 431, 436) are also the limits of the study. (i) Concerning the other modelling approaches mentioned, it is true that they require calibration with data from different physiological processes (e.g. leaf expansion, phenology) which are not easy to be measured. However, this effort is compensated by the fact that being the physiological processes at the basis of crop yield explicitly simulated (something completely overlook in the approach presented), it is possible to extend the analysis to conditions not explored with the field trials, like e.g. future climates or different contexts (management, climate).

- (ii) the fact that phenology is an input data, restrict the application of the analysis only to conditions explored with the field trials (as point above). So the methodology proposed in this study have clear value and clear advantages, but also clear limitations that should be discussed and mentioned as well in this chapter.

Reviewer #3 (Remarks to the Author):

General Comments

This paper combined three topics into a single paper: YpCGM, genomic prediction, and GWAS. The YpCGM method has considerable merit, but neither of the three topics were thoroughly studied. The YpCGM method was first reported in the Masuya and Shimono (2017) paper. The unique feature of the YpCGM method is replacing variety (or site) mean yield in the F-W method with climatic (or theoretical) potential yield. Since both methods rely on observed yield data, both are constrained by the extent of common shared varieties in multi-site and multi-year independent experiments. The paper could have included a comparison of regression results (yield ability and plasticity) between the F-W and YpCGM methods, providing an in-depth analysis on pros and cons of the two methods, and how lack of common shared varieties from multi-site and multi-year experiments might impact the estimated yield ability and plasticity.

CO₂ fertilization effect was not considered in the climate-driven crop growth model, some of the ‘genetic gain’ in yield ability can probably be attributed to yield gain from increasing CO₂ concentration.

The GWAS analysis lacks elaboration on identifying genes that could potentially contribute to yield ability or yield plasticity. GWAS analysis and GP on other yield contributing traits (e.g. days to heading, panicle number, and panicle length) was not included.

Most of the supplementary Figures 3 and 5-9 provide trivial details that could be greatly reduced.

Specific comments

L19: YpCGM  Define the acronym

L34: t/ha  Metric ton/ha?

L78-80: "However, the FW method can be used only with yield data measured at the same site in a side-by-side yield comparison, in which different cultivars grow together under identical environments"  This statement is flawed. The F-W method in the Finlay and Wilkinson (1963) paper and in the Masuya and Shimono (2017) paper had data from multiple years and multiple sites.

L80-81: "The plasticities of a cultivar determined under different environmental conditions are consequently not comparable over independent experiments"  This statement is flawed (see comment immediately above). As long as the same varieties are used, plasticity should be comparable over independent experiments in different environments. The problem occurs only if some varieties are not included in independent experiments in different environments (sites, years and management). The problem may depend on the extent of common shared varieties, which may deserve a separate treatment in another paper.

L96: observed Yp  What is observed Yp ?

L149: where the canopy is flooded  you mean the plants were flooded, not canopy?

L153-188: This section contains too much trivial details and can be greatly shortened

L230: satisfying $MAF \geq 0.5$  You mean 0.05 (i.e. 5%) for minor allele frequency?

L259-260: Z is the design matrix  Z is not in in equation

L260: Symbol epsilon is the vector of residuals  epsilon is not in equation

L317: Figure 1c and 1e  Y axis title should be lower Greek Alphabet beta not 'B'

L366: Figure 1d  Missing regression line for the 23 major cultivars

Reviewer #1

The author used a large historical rice data within a new simple crop growth model named YpCGM that involves only two critical genotype-specific parameters. The authors then applied genomic prediction and GWAS on the simulated parameters but their prediction accuracies were considerably low. I have a mixed feeling about this paper and I cannot say that I can accept it or rejected it before getting clarification from the authors on multiple issues stated below. Specifically, the description of the model is not very clear and it may not be practical to predict the performance of cultivars in unobserved environments as it requires some measured phenotypes as inputs. Moreover, based on what described in equation one, it may suffer from parameter equifinality.

> The YpCGM method is not a CGM *per se*, but rather is a method to extract two cultivar-specific parameters (α and β) from an analysis of the relationship between observed yield and predictions from a CGM (any CGM), and is therefore an extension of the Finlay and Wilkinson (1963) method. We describe this extension in lines 70-100. The CGM *per se* is not important; it represents only one example. Specifically, we applied this method to rice using a simple CGM (Masuya and Shimono, 2017). We have added an overview of the method in the text (L142-185) and in the Abstract (L27-31). We have also added a description of the CGM and its default parameters (Fig. S5).

Generally, the paper need to be fully restructured. Each section should contain what it supposed to include, e.g., introduction doesn't need to have a description of the method.

Introduction:

In general, the introduction is very long and some parts of it can be moved to the materials and methods (e.g. around the equation) or the results/discussion (the following paragraphs).

Line 104: CGM-WGP model can be applied to big data with limited number of phenotyped traits as well as limited number of phenotypic records per cultivar (see Jighly et al. 2022 <https://doi.org/10.1093/jxb/erac393>).

> We have restructured the Introduction (L55-109), and moved some parts to the Methods section (L143-159). We have also moved some parts of the Results section to the Discussion (L432-558).

Materials and methods:

It is not clear if YpCGM uses yield-ability and plasticity a single type only in equation 1 or

they are used in multiple equations along the whole model. If they are only using both of them in a single equation, then you will have a parameter equifinality problem

> We have added an explanation of the YpCGM method (L160-180), and a description of the CGM structure and its default parameters as Fig. S5.

Line 196–197: Did you considered the observed heading and maturity directly in the model or they were used to calibrate it? If the model assumes that heading and maturity should exist, then how can you predict the performance of the cultivar in unobserved environments (which is the most important application of crop growth models)? Another problem associate with that, if you are using the observed heading and maturity, these measures are usually correlated with yield. Therefore, it will be expected to get higher accuracy for yield prediction than other “more practical” models

> The observed heading and maturity dates are directly addressed in the model, as described in the Methods (L166-175). In terms of your second question, one of the strengths of our method is its high accuracy to calculate the yield ability and plasticity to allow researchers to combine large amounts of data from different studies. Different CGMs could be used for different species or purposes. (L520-558).

Line 201: If you are using the observed heading and maturity then of course the phenological parameters will not be required

> We don't use the phenology parameters in our analysis, but instead use them to run the model so that we can calculate *DVI* throughout the cropping season on the basis of the observed phenology (L166-175).

Line 209: I may misunderstand something here but if you have the measured SPY for all cultivars, why don't you use it instead of setting it to a single value of 8 t/ha

> *SPY* is set to a single value that represents a yield threshold for the majority of the cultivars. In this case, 8 t/ha represents a value achieved by 85% of the cultivars (L156-157).

Line 230: should be “< 0.5”

> We have corrected this as $MAF \geq 0.05$ (L235).

Line 247–250: g and p are generally highly correlated, so it will not be really useful to use them together.

> Yes, they are correlated, but they reflect different characteristics to some extent, the two analyses don't overlap completely, thus combining them in a single analysis can improve the result by accounting for those differences.. We therefore tested the possibility that combining them would improve the results, and our analysis confirmed this possibility (L344-353).

Line 264&265: We usually use the term "iteration" instead of cycles or samples

> We have corrected this to "iteration" (L267).

Line 296: any specific python library or command?

> We have added a description of all software packages and their settings in the Methods (L296-303).

Results

You need to present the correlation between yield-ability and plasticity with the observed phenotypes.

> We describe the relationship between the two parameters and the observed yield, panicle number, and panicle length in Fig. S9. Interestingly, both α and β were significantly positively correlated with panicle number, but significantly negatively correlated with panicle length. We describe these results in L337-342.

Line 305–309: Long sentence, please split

> We have asked our English editors to fix this and other writing problems for whole manuscript again.

Line 332: I would love to see how pBLUP and gBLUP will perform if applied directly to the observed phenotypes, e.g. yield.

> Although this would be interesting, it is beyond the scope of our study. We do, however, hope to study this in the future. In particular, we hope to improve pBLUP and gBLUP to calculate cultivar-specific values of β and α , and compare the results with the present YpCGM method. We discuss this briefly in the Discussion (L529-539).

Line 346–349: It might be better to describe the heritability then prediction accuracy

> We prefer to discuss accuracy first, since the method must first be accurate. Heritability then defines which genotypes will be most effective or practical to use in breeding.

Line 374: Large proportion of this heading can be moved to the discussion

> We have moved to the Discussion (L432-483).

Discussion

Large proportion of the discussion is just a repetition of what stated in other sections

> We have revised the Discussion to reduce this repetition.

Thank you for your efforts to improve our manuscript. We hope that our responses and the resulting replies will be satisfactory, but we will be happy to work with you to resolve any remaining issues.

Reviewer #2

GENERAL COMMENT:

This study proposes a methodology based on the integration of an empirical crop model and linear regression to better characterize rice cultivars in terms of yield potential and yield plasticity. The methodology is of interest, since it allows to use past yield and phenology data from multi-environmental trials to extract more information about GxE interactions for multiple cultivars. The dataset at the basis of the study is impressive, which enables the use of this empirical approach. The study is focused in Japan. The manuscript is well written and clear, although some work is needed in the introduction and in the discussion sections (see detailed comments below). The description of the methodology used is appropriate to ensure reproducibility.

That said, my main concern refers to the potential extension of the approach proposed to other crops (explicitly suggested at line 441–445) or to future conditions (which is suggested throughout the manuscript). I do not agree at all on this. As explained and mentioned many times in the detailed comments here below, the statistical approach proposed in this study has the clear advantage of requiring less efforts as compared to process-based modelling approaches (no calibration for specific sub-processes needed, like e.g., leaf expansion). However, this also determines clear limits. Since the physiological processes underlying crop yield are not understood, captured and explicitly simulated, this statistical approach cannot be extended to conditions different from those explored in the field trials used. That means not to future climates, where climate is expected to assume unprecedented features and patterns, nor to rice or other crops not grown under flooded conditions as in this study (where temperature and solar radiation are the only limiting factors for crop growth). The validity of this methodology to conditions where factors with non-linear effects on yield play a key role is also questionable (e.g., biotic/abiotic stressors, which have not been considered in this study, line 204–205), and need to be supported by dedicated studies.

So the study is of interest and might have clear value from a breeding perspective, but the authors should also clearly highlight the limits of the study and do not speculate about the extension of the methodology to conditions different from those explored with the field trials (future climates, other crops), since this is not supported by the results of this study at all. It can be mentioned in the conclusion as a perspective, as a hypothesis to be tested and validated with further studies.

Apologies in advance for reiterating this critical point many times in the detailed

comments here below, but in my opinion this is really a key issue of the manuscript.

> Although we recognize the validity of your criticisms, we believe that most result from a misunderstanding of what we did and our goals. First, our approach is NOT a purely “statistical approach” or a “physiological basis CGM approach”. Our use of a CGM is specifically chosen to allow researchers to extend our approach to conditions different from those explored in the original field trials; this is because a CGM integrates the model’s many mathematical descriptions of physiological processes and their responses to changes in the environment and management practices throughout the cropping cycle. It does not, as you note, provide insights into specific physiological processes, but that is not the point of the method or our study. In our approach, we used Y_p calculated by the CGM with fixed default parameters to reflect how the CGM accounts for the environment’s effects on rice yield. In fact, we found that Y_p is a good indicator for explaining variations in observed yield because its predictions are as good as those for the observed panicle number for each cultivar. This means that our Y_p can extract the cultivar-specific response to environments, without having to determine which underlying physiological processes are responsible for these responses. Thus, our approach can predict the potential response to future conditions so long as Y_p falls within the range of values actually experienced by cultivars.

In addition, the Y_p CGM method is intended to permit the use of data from many studies (i.e., big data) of a given crop species; we tested this concept in rice and found that two parameters could explain the variation of cultivar yield characteristics (yield ability and plasticity) by using Y_p calculated by the CGM. To apply this method to other crop species, researchers can use different crop models to calculate Y_p for their study species rather than our model. The analysis would also require different climate and management datasets. The goal is to standardize the yield ability of cultivars in response to Y_p under a given set of environmental conditions. So our method is applicable to other crop species so long as researchers understand that the values of α and β must be recalculated for each species or range of environmental conditions and that different models may perform better for different species.

SPECIFIC COMMENTS:

- Title: “weather-driven crop model”: weather variables are among the drive variables in all crop models. I strongly suggest to use/add “empirical “ because the model used in this study has a very strong empirical component.

> We prefer not to add “empirical” in the title because all crop models are, to a greater or

lesser extent, empirical. We do describe the “empirical” nature of our model in the Methods (L160).

- L96–98: please clarify how SPY is derived.

> In the methods, we clarify (L153-157) why we chose the value of 8 t/ha: because 85% of all cultivars (201 of the 237 cultivars) experienced at least this Y_p (Fig. S4).

- Introduction needs work. It seems more like a summary of the study rather than an analysis of the state of the art, of the context in which this study is placed. Please extend the analysis of the literature available beside the specific work of FW.

> We have revised the Introduction to improve its structure and content.

- L100. I do not agree. This sentence sounds like the parameterization for different physiological processes is something not actually needed and that it can be simply overlooked by using a big-data approach. Such parameterization, focused on multiple processes (e.g., leaf expansion, phenology, photosynthesis), is fundamental to avoid equifinality (i.e., same model output [in this case yield] achieved with different parameter sets), which can easily lead to loss of robustness when the crop model is applied to independent datasets.

> We apologize for our lack of clarity. We wanted to state that there are several ways to analyze data using a CGM. If enough data are available, a CGM can predict crop yield by integrating mathematical descriptions of physiological processes in response to changes in the environment and in management practices throughout the cropping cycle. The problem we attempted to solve is that there is often insufficient data to use that approach. Our proposed method allows the use of large, previously unused datasets by making them more compatible and, thus, easier to combine in a big data analysis. In specialized modeling studies, cultivar characteristics are expressed by cultivar-specific parameters in the mathematical equations that describe physiological processes, and the parameters are used for genomic analysis for yield (Chapman et al., 2003; Chenu et al., 2009; Jones et al., 2019), leaf expansion (Reymond et al., 2003), and flowering time (Nakagawa et al., 2005; Uptmoor et al., 2016). We describe this in more detail in the Discussion (L486-499).

- L104: As above: I do not agree. Sometimes it is better to have less data but of high quality rather than “big data” with no or very little understanding of the physiological processes

underlying the observed yield. The empirical, black box approach proposed in this study just needs observed yield and phenology, which for sure provides the advantage of requiring less effort as compared to the other modelling approaches mentioned (no calibration for specific processes needed). However, this also determines clear limits. Since the physiological processes underlying crop yield are not understood, captured and explicitly simulated, this statistical approach cannot be extended to conditions different from those explored in the field trials (e.g., future climates, different management contexts, different crops). I would suggest at least to the authors to discuss more this issue and clearly highlight the limits of the study in the discussion section (only advantages are mentioned). I would also better underline here the specific context of this study, with yield data coming from rice grown under optimal conditions (as explained at line 149–151) which allow to undertake a data-driven approach as that described in this study.

> We agree about the value of small, precise datasets. However, the goal of our method is to take advantage of the large available datasets when more precise data is not available. For example, no studies cover the large area of Japan that we covered and provide precise datasets for the whole study area. We describe this in L485-558.

- L110: heading date is not part of the “cultivation schedule”. It is phenological development of the crop, which should be entirely simulated, not provided as input. Cultivation schedule should refer only to sowing/transplanting, fertilization, weed and pest management, harvest events (maturity date instead is still phenological development as heading date). The fact that heading date is an input, is one of the reasons why the analysis cannot be extended to new environments (e.g., future climates).

> We have revised the text to correct this error (L171).

- L113–114: these are the pros, but there are also many cons like e.g., the impossibility to extend the analysis to conditions not explored with the field trials (see also previous comment).

> Please see our response above.

- L127: please delete reference to future climates. With the method described, prediction cannot be extended to conditions different from those explored while developing the linear model. To do so, process-based crop models need to be used.

> As noted earlier in our responses, we have added a note about the risks of extrapolating

beyond the range of values in the datasets we used. However, so long as an analysis occurs within that range of values, there's no reason why a properly calibrated CGM could not reliably predict future yield.

- L141: 20 is the minimum number of observations for each genotype, but I would suggest provide here also the maximum one. At least reference to any supplementary material where this information can be derived (Table S1). The number of observation for each genotype is a key information to evaluate the soundness of the analysis.

> We have added information in the Methods on the range of sample sizes (L127-129).

- L144: please clarify what 4.0 stands for or delete it.

> We have clarified the meaning of this number (L130-131).

- L204–206: the fact that heat, drought, and changes in CO₂ concentrations are not included in the study is exactly the reason why I suggest to the authors to delete from the manuscript any reference to extending prediction/support breeding for future climates. Those are expected to be key factors under future climates.

> As we noted above, we believe that our approach can be broadly valid so long as the prediction conditions fall within the range of conditions under which the yield data were measured. The problem then becomes the robustness of the underlying CGM. However, we have clarified that these factors were outside the scope of our study (L520-539).

- Line 209–211: the use of observed SPY highlights once more the strongly empirical nature of the approach used.

> As we note above, all modeling studies and GWAS studies rely on empirical measurement data.

- Line 335: prediction accuracy is pretty low. Please add to the discussion section a paragraph to discuss this issue and make comparison with available literature.

> We have added a description of this problem in the Discussion (L502-505).

- Line 375–388: please, clarify that you refer to the changing in climate observed in the past years (those explored with your dataset). The methodology presented in this study cannot support extending the analysis to future climates (an empirical model based on statistical

relationships does not support the extension to climate conditions not explored while developing the model). For this reason, I think that calling them “climate scenarios” can be misleading, because this term usually refers to future climate projections (SSPs×GCM combinations). If you want to keep using this term, please better explain what they are/how they have been derived.

> As noted in our previous responses, we have clarified that our approach should not be used to predict results beyond the range of conditions under which the field data were recorded, and that the quality of the results depends on the robustness of the underlying CGM under those conditions. Note that we used the 6342 field environments experienced by ‘Koshihikari’.

- Line 441–445: this is speculation not supported at all by the result of this study. In this study authors focused the analysis (i) on paddy rice grown under flooded conditions and optimal management (line 149–151), so without any limitations to crop growth other than temperature and solar radiation, and (ii) to a restricted area (temperate climate of Japan). According to the results of the study, it is impossible to say if this method would be valid under different conditions (e.g. rice grown in warmer climates with heat waves or water scarcity issues), let alone other crops not irrigated or irrigated but not at field capacity (as rice). So, the discussion should focus on comparing and discussing results with the literature available, not on speculating about applications of the methodology without being supported by the data presented. It can be reported as a perspective, to be tested and validated in dedicated future studies.

> We revised the Discussion to clarify that our method is valid only under a specific range of conditions, and that the yield calculations must be repeated using data for the conditions under which a new model will be applied; for example, if researchers change the focus from flooded rice (our study) to dryland rice, they would need to repeat our analysis using data from dryland conditions. In some cases, it may be necessary to choose an underlying CGM that is more effective for those conditions. We describe this in the Discussion (L485-558).

- Line 431–436: This manuscript propose a method based on empirical modelling and linear regression. For this reason, the “advantages” highlighted by the authors (line 431, 436) are also the limits of the study. (i) Concerning the other modelling approaches mentioned, it is true that they require calibration with data from different physiological processes (e.g. leaf expansion, phenology) which are not easy to be measured. However, this effort is

compensated by the fact that being the physiological processes at the basis of crop yield explicitly simulated (something completely overlook in the approach presented), it is possible to extend the analysis to conditions not explored with the field trials, like e.g. future climates or different contexts (management, climate).

(ii) the fact that phenology is an input data, restrict the application of the analysis only to conditions explored with the field trials (as point above). So the methodology proposed in this study have clear value and clear advantages, but also clear limitations that should be discussed and mentioned as well in this chapter.

> We added discussed these points in the Discussion (L485-558).

Thank you for your efforts to improve our manuscript. We hope that our responses and the resulting replies will be satisfactory, but we will be happy to work with you to resolve any remaining issues.

Reviewer #3

General Comments

This paper combined three topics into a single paper: YpCGM, genomic prediction, and GWAS. The YpCGM method has considerable merit, but neither of the three topics were thoroughly studied. The YpCGM method was first reported in the Masuya and Shimono (2017) paper. The unique feature of the YpCGM method is replacing variety (or site) mean yield in the F-W method with climatic (or theoretical) potential yield. Since both methods rely on observed yield data, both are constrained by the extent of common shared varieties in multi-site and multi-year independent experiments. The paper could have included a comparison of regression results (yield ability and plasticity) between the F-W and YpCGM methods, providing an in-depth analysis on pros and cons of the two methods, and how lack of common shared varieties from multi-site and multi-year experiments might impact the estimated yield ability and plasticity.

CO₂ fertilization effect was not considered in the climate-driven crop growth model, some of the ‘genetic gain’ in yield ability can probably be attributed to yield gain from increasing CO₂ concentration.

> We discuss these points in L468-473 and L485-558.

The GWAS analysis lacks elaboration on identifying genes that could potentially contribute to yield ability or yield plasticity. GWAS analysis and GP on other yield contributing traits (e.g. days to heading, panicle number, and panicle length) was not included.

> Our study focuses on describing the new method and illustrating how it can be applied.

Identifying specific genes is therefore beyond its scope. However, we have provided a description of potential genes identified by GWAS (L411-430) to show how our approach could be used in future research to identify specific genes of interest.

Most of the supplementary Figures 3 and 5–9 provide trivial details that could be greatly reduced.

> We believe that these supplemental figures are valuable because they clarify the methodology and provide useful data on the statistical distributions of the factors we studied. The whole point of supplemental figures is to provide data that are relevant and useful, with crucial details essential for understanding the paper. We want to keep the supplemental materials for these reasons.

Specific comments

L19: Y_p CGM  Define the acronym

> In highlight section of short space, it is difficult to acronym. We have revised the Highlight and the Abstract to clarify the meaning of the two components (Y_p and CGM).

L34: t/ha  Metric ton/ha?

> We did not change, SI units. "t/ha" is correct.

L78–80: “However, the FW method can be used only with yield data measured at the same site in a side-by-side yield comparison, in which different cultivars grow together under identical environments”  This statement is flawed. The F-W method in the Finlay and Wilkinson (1963) paper and in the Masuya and Shimono (2017) paper had data from multiple years and multiple sites.

> The FW method can be used only with yield data measured at the same site in a side-by-side yield comparison, in which different cultivars grow together under identical environments. The x -axis in the FW method represents the mean of all cultivars that were tested; to ensure that these data can be combined, the cultivars should experience identical growth environments to rule out differences that result from the environment rather than from characteristics of the cultivars. For example, if one experiment used 10 cultivars in Japan in 1950, and another experiment used 10 different cultivars in Egypt in 2022, the FW slopes of the two datasets cannot be combined as a single analysis for 20 cultivars; rather, the comparison would have to be limited to each set of 10 cultivars.

Note that Masuya and Shimono (2017) did not use the FW method, but instead ranked the yielding ability of three rice cultivars on the basis of Y_p . They compared the rank with the means of the observed yield of each cultivar from the data of all three cultivars grown at the same locations in the same years. Finlay and Wilkinson (1963) did only one FW analysis of the yield of 277 barley cultivars grown together at 3 locations \times 3 years, and did not compare FW results from other analyses.

L80–81: “The plasticities of a cultivar determined under different environmental conditions are consequently not comparable over independent experiments”  This statement is flawed (see comment immediately above). As long as the same varieties are used, plasticity should be comparable over independent experiments in different environments. The problem occurs

only if some varieties are not included in independent experiments in different environments (sites, years and management). The problem may depend on the extent of common shared varieties, which may deserve a separate treatment in another paper.

> Please see my previous response to your comment.

L96: observed Y_p  What is observed Y_p ?

> We apologize for the lack of clarity. We have revised the wording (L153-157).

L149: where the canopy is flooded  you mean the plants were flooded, not canopy?

> We have revised the description as you suggested (L135).

L153–188: This section contains too much trivial details and can be greatly shortened

> We have retained what we believe to be the most important details, particularly those that will help future researchers replicate our approach.

L230: satisfying $MAF \geq 0.5$  You mean 0.05 (i.e. 5%) for minor allele frequency?

> We have corrected this error (L235).

L259–260: Z is the design matrix  Z is not in in equation

> We have deleted our description of this variable.

L260: Symbol epsilon is the vector of residuals  epsilon is not in equation

> We have changed the variable name to ϵ in the equations and variable definitions

L317: Figure 1c and 1e  Y axis title should be lower Greek Alphabet beta not 'B'

> We have corrected the graph axes.

L366: Figure 1d  Missing regression line for the 23 major cultivars

> The regression results in this graph were not significant, so we did not provide a regression line. We have added “ns” after the r values to clarify this point.

Thank you for your efforts to improve our manuscript. We hope that our responses and the resulting replies will be satisfactory, but we will be happy to work with you to resolve any remaining issues.

Reviewers' comments:

Reviewer #1 (Remarks to the Author):

Thanks for your time responding to my comments. It is obvious that the described model suffers from parameter equifinality which is a serious problem the authors should improve. This is a common problem when multiple parameters are only used in a single equation (such as α and β in equation 1). Parameter equifinality is the reason why your parameters have high correlation with observed traits, high heritability but considerably low accuracy. Parameter equifinality usually leads to perfect results on the reference data that get vanished when applied for cultivars not included in the reference.

Regardless of the author's respond to my first comment in the previous round, they should test a scenario in which they simulate the phenology parameters so their model can be validated for practical implementation. They already have low prediction accuracy for the parameters, I'm wondering how much the accuracy can get further reduced using simulated phenology.

Reviewer #2 (Remarks to the Author):

Thanks for revising the manuscript and addressing the points of concerns that were raised. Most of my doubts have been solved. The introduction looks pretty much the same of the earlier version but the manuscript is now clearer, and the limits of the approach proposed are now at least mentioned, in particular the risk of extrapolating beyond the conditions explored with the field trials (cited at the end of the conclusion section).

Reviewer #3 (Remarks to the Author):

The authors have largely addressed reviewers' comments. But I still believe the statement in lines 80-82 "the FW method can only be used to compare cultivars with yield data measured at the same site in a side-by-side yield comparison, in which different cultivars grow together under identical environments" is flawed. Fig, 2 in the Finlay and Wilkinson (1963) paper includes results from three different sites with each site having data for three seasons. My understanding of the FW method is that it can be used to compare cultivars with yield data measured at different sites and different seasons as long as the same cultivars were used in each site \times season combination. A very valuable contribution of the paper (or a new paper) could be to explore the potential of using the FW method to compare cultivars with yield data measured at different sites and different seasons with partially overlapping cultivars between sites and seasons.

Reviewer #1

Thanks for your time responding to my comments. It is obvious that the described model suffers from parameter equifinality which is a serious problem the authors should improve. This is a common problem when multiple parameters are only used in a single equation (such as α and β in equation 1). Parameter equifinality is the reason why your parameters have high correlation with observed traits, high heritability but considerably low accuracy. Parameter equifinality usually leads to perfect results on the reference data that get vanished when applied for cultivars not included in the reference.

>Thank you for your comments on the robustness of the regression model. As you pointed out, in the absence of parameter identifiability, different models (here we consider models with different estimates as different models) fit the data well (equifinality), but prediction becomes more difficult. We validated the regression model for all cultivars with leave-one-environment-out cross-validation and compared the goodness of fit of the model (high coefficient of determination) and the predictive ability of the model (high coefficient of determination for prediction) (Supplemental Fig.S9, newly added) (M&M L186-194, Result L356-364). The results showed that the differences between the two coefficients of determination were small, confirming the robustness of the model.

It is known that correlations between regression coefficients always occur in regression models. $y = b + \alpha x$, where the correlation between b and α is negative when the mean of x is greater than 0, the correlation between b and α is positive when the mean of x is less than 0, and the correlation between b and α is zero when the mean of x is 0. In this study, the correlation between the two (here β and α) is smaller but not zero (Fig.1e) because the intercept, beta, at $x = \text{SPY}$ (in equation, $y = \beta + \alpha (x - \text{SPY})$) was used instead of b of intercept at $x = 0$. In order to consider the correlation between the coefficients in the genomic prediction modeling, we also constructed a prediction model using a multiple trait model. As a result, the prediction accuracy was further improved, especially for β (Fig.2, revised from previous version) (M&M L265-306, Results L373-383). The genomic prediction accuracy for β is not lower than the prediction accuracy reported for the usual quantitative traits.

Regardless of the author's respond to my first comment in the previous round, they should test a scenario in which they simulate the phenology parameters so their model can be validated for practical implementation. They already have low prediction accuracy for the parameters, I'm wondering how much the accuracy can get further reduced using simulated phenology.

>In this study, regressions models were constructed for cultivars based on historical breeding data, and genomic predictions and GWAS of the parameters were performed. We evaluated the models as described above because we did not fully understand what kind of analysis the simulation of phenology parameters refers to, and because we believed that the robustness of the models could be verified by the cross-validation. In some cultivars, the coefficient of determination for prediction was lower than the coefficient of determination for fitting, but the number of data used in the regression was small for all these cultivars (Supplemental Fig.S9) (M&M L186-194, Result L356-364). This suggests that, as the number of data increases, the robustness of the regression model is expected to increase further. The non-robustness of the model, especially equifinality and identifiability, is not considered to be an issue, and there are no practical problems with the modeling method.

Reviewer #2:

Thanks for revising the manuscript and addressing the points of concerns that were raised. Most of my doubts have been solved. The introduction looks pretty much the same of the earlier version but the manuscript is now clearer, and the limits of the approach proposed are now at least mentioned, in particular the risk of extrapolating beyond the conditions explored with the field trials (cited at the end of the conclusion section).

>Thank you for your suggestions during this review processes.

Reviewer #3:

The authors have largely addressed reviewers' comments. But I still believe the statement in lines 80-82 "the FW method can only be used to compare cultivars with yield data measured at the same site in a side-by-side yield comparison, in which different cultivars grow together under identical environments" is flawed. Fig. 2 in the Finlay and Wilkinson (1963) paper includes results from three different sites with each site having data for three seasons. My understanding of the FW method is that it can be used to compare cultivars with yield data measured at different sites and different seasons as long as the same cultivars were used in each site \times season combination. A very valuable contribution of the paper (or a new paper) could be to explore the potential of using the FW method to compare cultivars with yield data measured at different sites and different seasons with partially overlapping cultivars between sites and seasons.

>We revised the manuscript in Introduction (L81-84).

REVIEWERS' COMMENTS:

Reviewer #1 (Remarks to the Author):

Thanks for your response and I am really sorry for the delay in reviewing the manuscript due to unforeseen personal circumstances.

The main concern for me is still the use of actual phenology traits are not available when predicting the performance in untested environments or future climates. There are several models that can predict phenology traits (e.g. Christy B, Berger J, Zhang H, Riffkin P, Merry A, Weeks A, McLean T, O'Leary GJ. 2019. Potential yield benefits from increased vernalisation requirement of canola in Southern Australia. *Field Crops Research* 239, 82–91.). Such models can be used to predict phenology traits, then you can use your model to test the prediction accuracy using the simulated phenology traits instead of the actual ones (as stated in line 164).

Anyway, I believe that the paper has its own novelty in the current form. Running this scenario will strengthen the practical applications of your work. If the authors have difficulties to run the analysis, they could add it as a discussion point that using the actual phenological traits could have led to over estimate the prediction accuracy but further research is needed to confirm that.

Reviewer #1

Thanks for your response and I am really sorry for the delay in reviewing the manuscript due to unforeseen personal circumstances.

The main concern for me is still the use of actual phenology traits are not available when predicting the performance in untested environments or future climates. There are several models that can predict phenology traits (e.g. Christy B, Berger J, Zhang H, Riffkin P, Merry A, Weeks A, McLean T, O'Leary GJ. 2019. Potential yield benefits from increased vernalisation requirement of canola in Southern Australia. *Field Crops Research* 239, 82–91.). Such models can be used to predict phenology traits, then you can use your model to test the prediction accuracy using the simulated phenology traits instead of the actual ones (as stated in line 164).

Anyway, I believe that the paper has its own novelty in the current form. Running this scenario will strengthen the practical applications of your work. If the authors have difficulties to run the analysis, they could add it as a discussion point that using the actual phenological traits could have led to over estimate the prediction accuracy but further research is needed to confirm that.

>Thank you for your comments. We added the point in Discussion (L544-549).